# Bionic Design and Adsorption Performance Analysis of Vacuum Suckers

**DOI:** 10.3390/biomimetics9100623

**Published:** 2024-10-14

**Authors:** Peng Xi, Yanqi Qiao, Xiaoyu Nie, Qian Cong

**Affiliations:** 1College of Agricultural Engineering, Shanxi Agricultural University, Jinzhong 030801, China; xipeng@sxau.edu.cn (P.X.); 20233053@stu.sxau.edu.cn (Y.Q.); http2571442822@163.com (X.N.); 2Dryland Farm Machinery Key Technology and Equipment Key Laboratory of Shanxi Province, Shanxi Agricultural University, Jinzhong 030801, China; 3College of Biological and Agricultural Engineering, Jilin University, Changchun 130022, China; 4Key Laboratory of Bionic Engineering Ministry of Education, Jilin University, Changchun 130022, China

**Keywords:** engineering bionics, sucker, abalone, sealing ring, adsorption

## Abstract

This study addresses the problem that the traditional method is not effective in improving the adsorption performance of vacuum suckers. From the perspective of bionics, the adsorption performance of bionic suckers based on the excellent adsorption of the abalone abdominal foot was studied. A bionic sucker was designed by extracting the sealing ring structure of the abdominal foot tentacle. The bionic sucker was subjected to tensile experiments using an orthogonal experimental design, and the adsorption of the bionic sucker was simulated and analyzed to explore its adsorption mechanism. The results show that the primary and secondary factors affecting the adsorption of the sucker are the number of sealing rings, the width of sealing rings and the spacing of sealing rings. At 60% vacuum, the bionic sucker with two sealing rings, a 1.5 mm sealing ring width and 3 mm sealing ring spacing has the largest adsorption force, and its maximum adsorption force is 15.8% higher than that of the standard sucker. This study shows that the bionic sucker design can effectively improve the adsorption performance of the sucker. The bionic sucker had a different stress distribution on the sucker bottom, which resulted in greater Mises stress in the sealing ring and the surrounding area, while the Mises stress in the central area of the sucker was smaller.

## 1. Introduction

As important terminal actuators in vacuum adsorption devices, vacuum suckers have the advantages of high work efficiency, many application scenarios, low cost and no pollution. Therefore, they have been widely used in industrial and civil fields to meet increasingly diverse needs [1,2,3]. When a vacuum sucker is working, a certain degree of vacuum is generated in the closed space between the sucker and the adsorption surface by excluding the gas between the sucker and the adsorption surface, so that the sucker is firmly adsorbed on the surface of the object through atmospheric pressure [4,5]. However, vacuum suckers are prone to problems such as poor sealing of the sucker and vacuum leakage during adsorption, which leads to a decrease in the adsorption force of the sucker and even adsorption failure, resulting in an increase in production costs and serious safety concerns [6,7]. Therefore, in order to improve the adsorption capacity of vacuum suckers and meet the higher functional requirements for vacuum suckers in all fields, it is of great significance to develop vacuum suckers with strong adsorption capacity, good stability and high compatibility with adsorption surfaces.

To date, domestic and foreign scholars and related engineers and technicians have carried out comprehensive and in-depth research on improving the adsorption performance of vacuum suckers. The research focuses mainly on two aspects, namely, sucker structure optimization and sucker material improvement. The structural optimization of the sucker is mainly based on the use environment of the sucker. For example, the corrugated pipe type sucker with sealed lip designed by the Schmalz Company in Germany can adapt to the adsorption of different food packaging bags, and can also be firmly grasped under the high-speed movement of the sucker [8]. The sucker with a sponge at the bottom designed by Aribest is suitable for objects with very rough or uneven adsorption surfaces [9]. At present, vacuum suckers are mainly divided into the standard flat type, deep concave type, sponge type, short bellows type, long bellows type, elliptical type, thin object type, elliptical concave type, nozzle type and so on. These different structures are suitable for different working environments and adsorption surfaces, such as flat, curved, rough, slender tubular and other surfaces. The sucker material is mainly selected according to the surface roughness, ambient temperature, oil resistance and other conditions of use. At present, there are diverse vacuum suckers with differences in structure, but the basic structure has been finalized. Therefore, it is difficult to carry out novel innovations in structure to improve the adsorption capacity of suckers. The selection of vacuum sucker materials is mainly based on the requirements of the actual use environment. There are many limiting factors for improving the adsorption performance by changing the sucker material, and the actual effects are not obvious. In order to further improve the adsorption performance of vacuum suckers to meet the development needs of industry, the theory of engineering bionics has become a new approach to improving their adsorption performance.

After a long period of evolution, organisms have formed their own unique abilities to adapt to the natural environment. Among them, adsorption is one of the abilities of organisms. Through adsorption, animals can carry out various activities such as crawling, predation, grabbing, escaping, etc., thus protecting themselves and the survival and reproduction of populations [10,11,12]. Common organisms with adsorption capacity include octopus, leech, abalone, clingfish and remora. The suckers on these organisms have strong adsorption capacity, and can not only adsorb on smooth surfaces, but also have good adsorption capacity on non-smooth surfaces [13,14,15,16]. In order to improve the adsorption performance of the vacuum sucker, relevant researchers have used engineering bionics methods to design the sucker to improve its adsorption capacity. Francesca Tramacere et al. established a three-dimensional model of the octopus sucker by means of ultrasonic and nuclear magnetic imaging, and prepared an imitation octopus sucker entity using silicone materials. It can be seen from their experiment that the adsorption force of the imitation octopus sucker with a diameter of 2 cm was about 8 N, which results in good adsorption capacity [17]. Petra Ditsche et al. designed a bionic sucker with a soft and hard double-layer structure based on the sucker structure of the Northern clingfish, which has good adsorption effect on both smooth and rough surfaces [18,19]. Ding et al. designed a bionic sucker with multiple folds and an array of small holes based on the pore cavity structure of a leech sucker. It can be seen from their experimental analysis that these suckers have the advantages of large Mises stress, good sealing properties, a strong adsorption force and good shear resistance [20]. Based on the study of octopus suckers and muscles, Greco et al. developed a flexible integrated bionic sucker for wet conditions, which achieved an adsorption pressure of 6 kPa in 300 ms and had good adsorption effects [21]. Cong et al. designed and processed the pit shape on the bottom surface of a sucker imitating a leech sucker, so that the working surface of the sucker formed a number of smaller multi-dimensional suckers. Through experiments and finite element analysis, it could be seen that the existence of pit structures on the working surface of the bionic sucker significantly increased its adsorption force. Compared with the average adsorption force of the standard sucker on the substrate surface, the maximum and minimum growth rates of the average adsorption force of the bionic sucker were 49.21% and 14.00%, respectively [22]. By further observing the structure of the octopus sucker, Kim et al. used a flexible film to separate the vacuum cavity between the suckers, so that the leakage of a single sucker in the sucker array only affected the leaking sucker, which greatly improved the stability of the sucker adsorption [23]. Wang et al. have developed remora-like suckers that can achieve remora-like ‘hitchhiking’ capabilities by generating considerable shear resistance through carbon fiber spines inside the sucker [24].

At present, there are many studies on the bionic design of suckers using octopus, leech, clingfish and other adsorption organisms as bionic prototypes, but there are few studies on the design of bionic suckers with abalone as a bionic prototype Abalone is an adhesive organism in the ocean whose abdominal foot has strong adhesion capabilities [25,26,27]. According to reports, an abalone with a body length of about 15 cm has an adhesion force of up to 200 kg, highlighting the organism’s strong adhesion force [28]. Due to the strong adhesive properties of abalone, researchers have conducted extensive studies on the adhesion of the organism’s muscular foot. Lin et al. studied the American red abalone and found that its abdominal foot surface is composed of fibers with two sizes. This multi-level fiber structure enables the abdominal foot sucker to form an interlocking structure on surfaces with a variety of roughness types, effectively improving the adaptability of abalone to different adhesion surfaces [29]. Li et al. tested the adhesion force of abalone in both water and air using various force measuring plates. The authors found that the adhesion force of the abalone’s abdominal foot primarily comes from vacuum adhesion forces, van der Waals forces, and capillary forces [30]. Xi analyzed measurements of abalone’s adhesion force on different force measuring plates and determined that the vacuum adhesion force plays a significant role in the total adhesion force of abalone [31].

Based on the good adsorption and adaptability of abalone on different morphological surfaces, this paper selects abalone with good adsorption as a bionic prototype from the perspective of bionics. The bionic vacuum sucker is designed by extracting the surface morphology of an abalone gastropod, and the designed bionic sucker is processed and prepared. According to the method of experimental optimization design, adsorption tests of standard and bionic suckers were carried out, and the influence of bionic structural parameters on the adsorption of suckers was analyzed. The finite element method was used to analyze the force on the bottom surface of the sucker, and the mechanism of the high adsorption of the bionic sucker was analyzed based on the adsorption results of the sucker.

## 2. Materials and Methods

### 2.1. Observation of Abalone Abdominal Foot

The abalone used in the experiment is Haliotis discus hannai, purchased from an aquatic market. The mass of abalone used was in the range 55–65 g [32,33]. Abalone comprise the surface shell and the internal gastropod, as shown in Figure 1a–c. The state of abalone gastropod adsorption is shown in Figure 1d. The abalone gastropod was observed by a stereomicroscope (Stemi 2000-C, ZEISS, Oberkochen, Germany), and it was found that the gastropod was surrounded by tentacles, as shown by the red dotted line in Figure 1d. When the abalone is pulled upward, the circular abdominal foot tentacles will quickly shrink inward and squeeze each other, forming a sealing ring structure to improve the sealing performance of the abdominal foot sucker, thereby improving the adsorption capacity of the abalone. In this paper, the sealing ring structure formed by abalone tentacles is extracted and used as a design feature of the bionic sucker.

### 2.2. Bionic Sucker Design

In order to carry out bionic design on the bottom surface of the vacuum sucker, the flat sucker commonly used in the industry was selected as the standard sucker. The diameter of the sucker is 60 mm. The structural parameters of the standard sucker were as shown in Figure 2. Using the structural parameters of the standard sucker, a three-dimensional model of the sucker was established, as shown in Figure 3.

According to the structural size of the standard sucker and based on the structural characteristics of the sealing ring formed by the tentacle, the bionic design of the standard sucker was carried out. The width, number and spacing of the sealing rings were selected as the design feature factors of the bionic sucker.

Based on the characteristics of the sealing ring formed by the contraction of the abdominal foot tentacles during abalone adsorption, the bionic design of the standard sucker was carried out. Because the width and number of sealing rings affect the contact pressure and sealing between the sucker and the adsorption surface, and because the spacing of the sealing rings affects the stress distribution on the bottom surface of the sucker, the width, number and the spacing of the sealing rings were selected as the characteristic design factors of the bionic sucker. Depending on the size of the bottom surface of the standard sucker and the width of the abdominal foot tentacles, reasonable values for these three characteristics (width, number and spacing of the sealing rings) were selected. Three levels were selected for each factor. The widths of the sealing rings (D) were 1.5 mm, 2.5 mm and 3.5 mm; the numbers of sealing rings (N) were 1, 2 and 3; and the sealing ring spacings (L) were 2 mm, 3 mm and 4 mm. The characteristics of the bionic sucker are shown in Table 1. The three-dimensional model of the bionic sucker is shown in Figure 4. The height of the sealing ring is designed to be 0.3 mm, and the outermost sealing ring is 4 mm away from the edge of the sucker. When designing the bionic sucker, the sealing ring is first arranged from the outer edge of the sucker. The height of the sealing ring is designed to be 0.3 mm. If the height is too large, only the sealing ring will contact the adsorption surface when the sucker is adsorbed, and other areas of the sucker have difficulty attaching to the adsorption surface. If the height is too small, the sealing ring will fail or cannot play a role.

### 2.3. Adsorption Experiment of Suckers

#### 2.3.1. Preparation of Sucker Samples

In this paper, the standard and bionic suckers were prepared by mold pouring. The specific process of preparing the sucker was as follows: (1) According to the structural parameters of the standard and bionic suckers, the mold for pouring the sucker was designed. The mold was divided into an upper mold and lower mold; (2) The designed sucker mold was processed and prepared by 3D printing. (3) A layer of vaseline was coated on the surface of the inner cavity formed by the mold to facilitate the forming and demolding of the sucker. (4) The mixture of silica gel and stationary liquid was prepared according to the ratio of 100:2 and fully stirred evenly. (5) The mixture was poured into the cavity formed by the upper mold and the lower mold, and a 1 kg weight was placed on top of the upper mold to facilitate the extrusion of the excess mixture in the cavity. (6) The sucker mold was left standing for 3–4 h and was removed when the mixture had solidified to obtain the sucker. The sucker mold is shown in Figure 5, and the prepared suckers are shown in Figure 6.

#### 2.3.2. Sucker Tensile Test

The tensile testing machine used in this experiment was the HLD-500N (HANDPI, Wenzhou, China) device, and a smooth glass plate was selected as the adsorption bottom surface for the experiment. In the experiment, the glass plate was first fixed on the base of the tensile testing machine, and the sucker was connected with the hook of the testing machine. Then, the test machine was slowly lowered until the bottom of the sucker was in contact with the glass plate, and a vertical downward pressure of 20 N was applied at the top of the measured sucker so that the bottom of sucker was completely attached to the glass plate, the air between them is discharged and the sucker was adsorbed to the glass plate. Then, the tensile testing machine was started and raised until the sucker was completely separated from the glass plate. The tensile testing machine is shown in Figure 7a. The tensile force of the sucker during the whole experiment process was recorded by the testing machine and is shown in Figure 7b.

It can be seen from Figure 7b that when the experiment began, with the gradual upward lifting of the tensile testing machine, the pull force of the vacuum sucker rapidly increased, and the adsorption force curve rose rapidly until the adsorption force between the sucker and the glass plate could not resist the upward pull of the sucker, and the sucker was detached from the glass plate. At this time, the adsorption force curve of the sucker rapidly decreased. As the sucker was contracted and deformed by the upward pulling force, the edge of the sucker formed a sealing structure again with the glass plate to resist the upward pulling force, and the adsorption force curve in Figure 7b increased briefly again. When the sucker completely separated from the glass plate, the adsorption force curve of the sucker decreased directly to zero. The maximum value in the tensile test was used as the adsorption force of the sucker on the glass plate. In the experiment, five tensile tests were performed on each sucker, and the average value of the five test results was taken as the final result.

## 3. Results

### 3.1. Orthogonal Experimental Design

According to the design characteristics of the three factors and three levels of the bionic sucker, the L_9_(3^4^) level table was selected for the preparation of the test plan. Regardless of the interaction between the factors, the No.10 group was used as the control group to select the standard sucker. The orthogonal test scheme of the suckers and the range analysis of the test results are shown in Table 2.

### 3.2. Analysis of Orthogonal Test Results

According to the test results in Table 2, the adsorption forces of the No.2, No.3 and No.5 bionic suckers were greater than that of the No.10 standard sucker, and the adsorption forces of the other bionic suckers were less than that of the No.10 standard sucker, while the adsorption force of No.2 bionic sucker was the largest, as shown in Figure 8. The range analysis method was used to analyze the test results. It was determined that the primary and secondary factors affecting the adsorption force of the bionic sealing rings were the number of sealing rings (N), the width of the sealing rings (D) and the spacing of the sealing rings (L). The optimal combination of the test was N_2_D_1_L_2_, corresponding to the structural parameters of the No.2 bionic sucker. The adsorption force of the No.2 bionic sucker was 15.8% higher than that of the No.10 standard sucker. In order to further verify the test results of the optimal combination and estimate the test error, the No.2 bionic sucker with the optimal combination parameters was repeatedly tested under the same test conditions, and the square sum of test error deviation Se and its degree of freedom f_e_ were calculated, as shown in Table 3. The repeated tests showed that the average adsorption force of the optimal combination of bionic No.2 sucker was 61.8 N, and the sum of squared errors was 0.187. The test had good repeatability.

The No.2 bionic sucker had the largest adsorption force, and the structural parameters of the sealing rings with the greatest influence on the adsorption force were the number of sealing rings (N) and the width of the sealing rings (D). It can be seen from Table 2 that when the sealing ring width (D) was 1.5 mm, 2.5 mm and 3.5 mm, the No.2, No.5 and No.8 bionic suckers with two sealing rings had the largest adsorption force among the No.1–3, No.4–6 and No.7–9 bionic suckers, respectively. The bionic suckers with two sealing rings had a relatively large adsorption force, indicating that one sealing ring is not sufficient to achieve the best sealing effect between the sucker and the adsorption surface. When there are three sealing rings, too many sealing rings are in contact with the adsorption surface, which disperses the contact pressure of the sealing rings, thus reducing the adsorption effect of the sucker. Sealing ring width (D) also plays an important role in the adsorption of suckers. The No.1, No.2 and No.3 bionic suckers had the largest adsorption force among the bionic suckers with 1, 2 and 3 sealing rings, respectively. The results show that the smaller the sealing ring width, the better the adsorption effect. When the sealing ring width was 1.5 mm, the contact pressure between the sealing ring and the adsorption surface was the largest, and the friction force of the sucker edge was greater when resisting the upward pull of the sucker, resulting in the best sealing effect. Therefore, the No.2 bionic sucker had the best adsorption effect.

### 3.3. Simulation Analysis of Sucker Adsorption

In order to explore the mechanism of the bionic sealing ring structure on the adsorption performance of suckers, the finite element simulation method was used. The finite element analysis software ansysworkbench16.0 was used for simulation analysis. The force measuring plate used in the adsorption test was a glass plate with a density of 2.46 g/cm^3^ and an elastic modulus of 68.9 GPa. The material of the vacuum sucker was rubber, which is a hyperelastic material. The Mooney−Rivlin 2 Parameter model was selected in the material library. Since the adsorption of a sucker is a type of contact problem, the force measuring plate was set as the target surface, and the vacuum sucker was set as the contact surface. The force measuring plate was set as a fixed constraint, and then a certain load was loaded on the top of the sucker. In this paper, the internal vacuum degree was 60% when the vacuum sucker was adsorbed, and the 60% standard atmospheric pressure, which was 60,795 Pa, was loaded vertically downward on the upper surface of the sucker. The loaded pressures are shown in Figure 9. The parameter setting of the bionic sucker is consistent with that of the standard sucker.

### 3.4. Adsorption Mechanism Analysis of Sucker

Based on the finite element simulation, the Mises stress diagram of the bottom surface of the sucker was obtained. The No.2 bionic sucker with the best adsorption in the tensile test, the No.4 bionic sucker with the worst adsorption and the standard sucker were selected for comparative analysis. In the Mises stress diagram of the simulation results, the Mises stress in different regions was measured from the edge to the center along the radial direction of the sucker bottom through the probe function. Among them, the Mises stress of the No.2 and No.4 bionic sucker was selected in six regions, as shown in Figure 10a,b (The six regions of the No.2 bionic sucker were the outer edge of the outer sealing ring of the sucker, the outer sealing ring of the sucker, the middle area of the two sealing rings, the inner sealing ring, and the inner deep blue center area of the sucker, respectively. The six areas of the No.4 bionic sucker were the outer edge of the sucker sealing ring, the sealing ring, and the deep blue area inside the sealing ring, respectively). The Mises stress levels of seven areas in the standard sucker were selected at the corresponding positions on the No.2 and No.4 bionic suckers, as shown in Figure 10c. By comparing the Mises stress values of the bottom surface of the No.2 and No.4 bionic suckers in Figure 9 with the standard sucker, it can be seen that the Mises stress on the bottom surface of the bionic sucker is very different from that of the standard sucker. Compared with the standard sucker, the Mises stress value of the bionic sucker on the sealing ring and its surrounding area is larger, while the Mises stress value in the deep blue area in the center of the sucker is smaller.

The Mises stress (50,548 Pa, 58,226 Pa, 39,544 Pa, 46,848 Pa) of the No.2 bionic sucker on the two sealing rings and the surrounding areas on both sides are greater than the Mises stress (38,488 Pa, 23,827 Pa, 16,531 Pa, 12,076 Pa) in the same area of the standard sucker, as shown in Figure 11a. The No.4 bionic sucker has the same characteristics, as shown in Figure 11b. The Mises stress values of the No.2 and No.4 bionic suckers in the center dark blue area are slightly lower than those of the standard suckers in the same area, as shown in Figure 11c,d.

Comparing the Mises stress values of the No.2 and No.4 bionic suckers, it can be seen that the Mises stress of the No.4 bionic sucker on the sealing ring and the surrounding area is greater, while the Mises stress of No.2 bionic sucker is relatively small, as shown in Figure 11e. In the deep blue area of the center of the sucker, the Mises stress values of the two are basically equal, as shown in Figure 11f. These results demonstrate that the No.2 bionic sucker has better adsorption than the No.4 bionic sucker.

It can be seen that the bionic design of the vacuum sucker changes the stress distribution on the bottom surface of the sucker. Compared with the standard sucker, the Mises stress value on the sealing ring and the surrounding area is larger, while the Mises stress value in the central area of the sucker is smaller. However, the increase in Mises stress in the sealing ring area of the sucker is greater than the decrease in Mises stress in the central area of the sucker, so that the adsorption of the No.2 bionic sucker is greater than that of the standard sucker.

## 4. Conclusions

(1)The bionic design of the vacuum sucker can effectively improve the adsorption performance of the sucker. The primary and secondary factors affecting the adsorption performance of the sucker are the number, width and spacing of the sealing rings.(2)From the adsorption experiments of the sucker, it can be seen that at 60% vacuum, the bionic sucker with two sealing rings, a sealing ring width of 1.5 mm, and sealing ring spacing of 3 mm has the largest adsorption force. The maximum adsorption force is 15.8% higher than the standard sucker.(3)Compared with the standard sucker, the bionic sucker has a larger Mises stress on the sealing ring and the surrounding area, while the Mises stress in the central area of the sucker is smaller.

In this paper, the adsorption performance of vacuum suckers is improved by bionic methods, and the adsorption mechanism of the bionic sucker is analyzed. However, based on the various sizes and types of vacuum suckers and the different surface shapes required for adsorption, the bionic shape and size parameters that need to be matched and adapted are also different. Therefore, improving the application range of bionic shape size and establishing selection principles for bionic shape size for different suckers that can be applied in different production environments, need to be further studied.

## Figures and Tables

**Figure 1 biomimetics-09-00623-f001:**
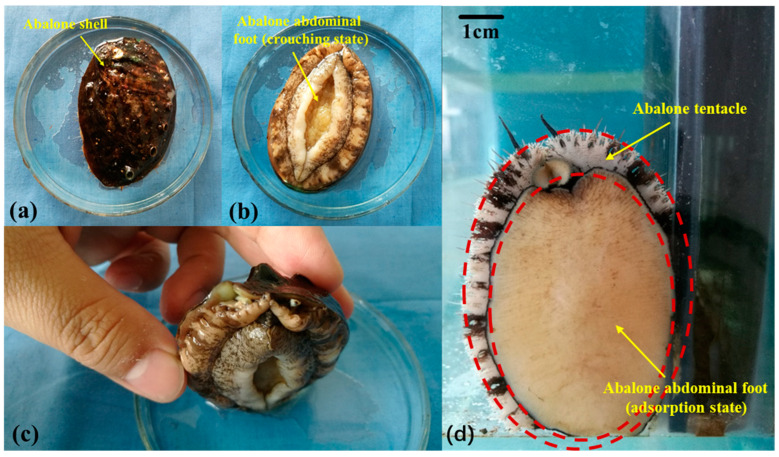
(**a**) Abalone shell; (**b**) abalone abdominal foot (crouching state); (**c**) the positional relationship between the abdominal foot and the shell; (**d**) abalone abdominal foot surface and tentacle.

**Figure 2 biomimetics-09-00623-f002:**
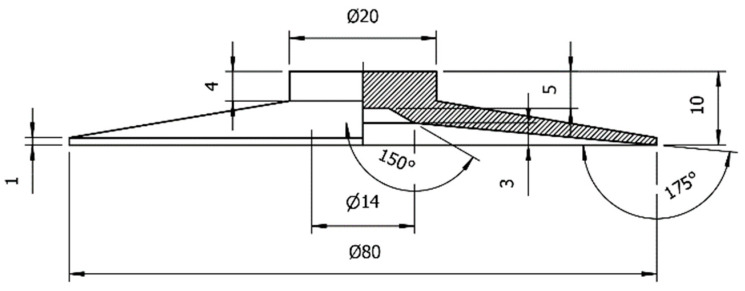
Structural parameters of the standard sucker.

**Figure 3 biomimetics-09-00623-f003:**
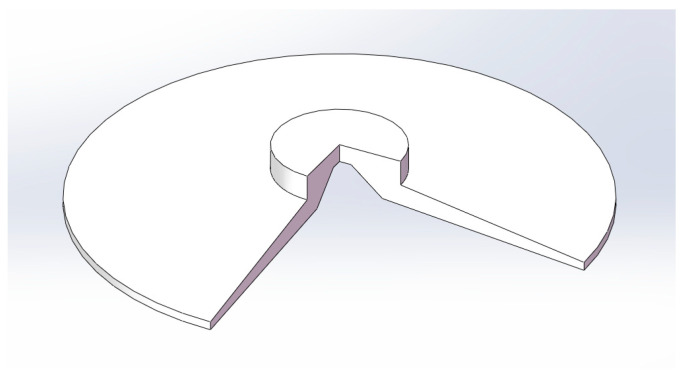
**A** 3D model of the standard sucker (part sectioned view).

**Figure 4 biomimetics-09-00623-f004:**
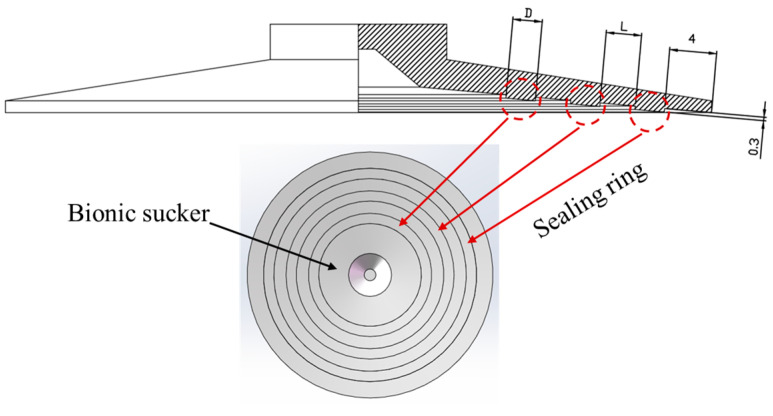
Design drawing and three-dimensional model of bionic sucker.

**Figure 5 biomimetics-09-00623-f005:**
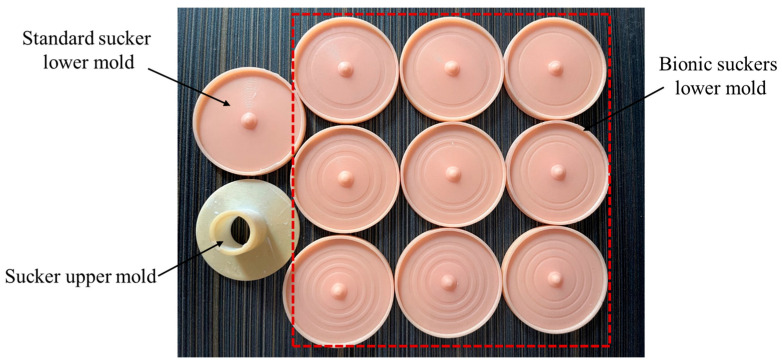
The standard and bionic suckers; upper and lower molds.

**Figure 6 biomimetics-09-00623-f006:**
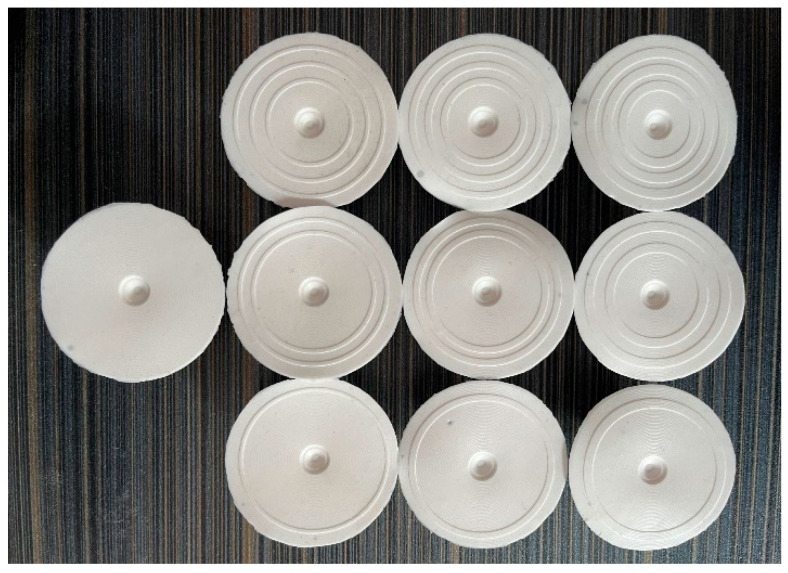
The standard and bionic suckers.

**Figure 7 biomimetics-09-00623-f007:**
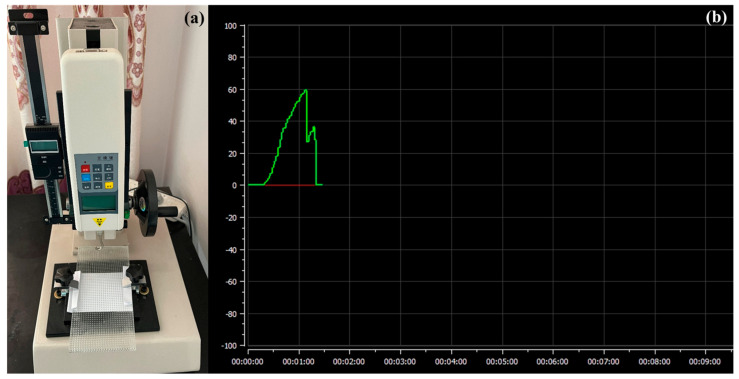
(**a**) Tensile test bench; (**b**) the sucker adsorption force curve.

**Figure 8 biomimetics-09-00623-f008:**
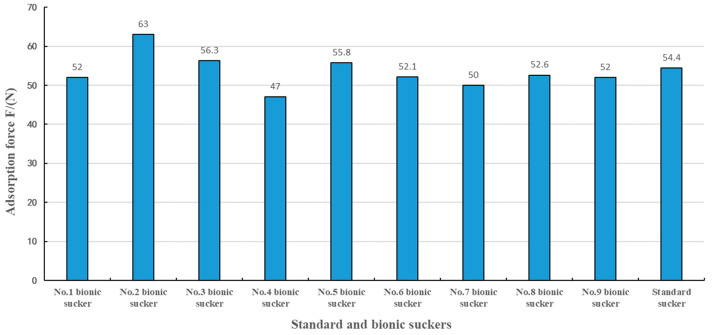
The adsorption force of the No.1 to No.9 bionic suckers and the standard sucker.

**Figure 9 biomimetics-09-00623-f009:**
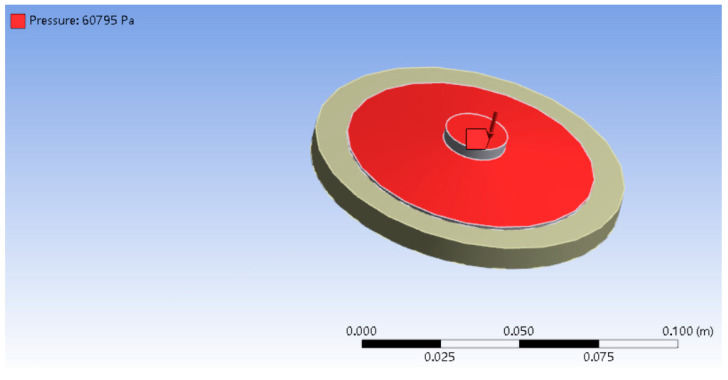
The top surface loaded pressure on the sucker in finite element analysis.

**Figure 10 biomimetics-09-00623-f010:**
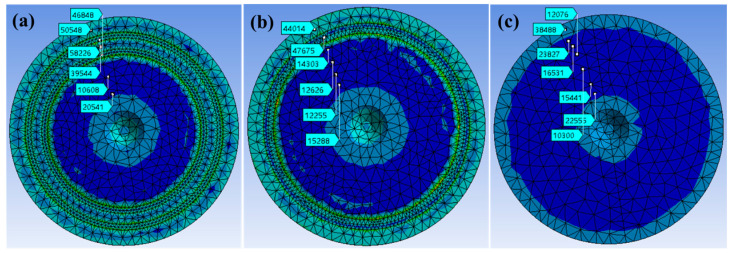
(**a**) The Mises stress of No.2 bionic sucker; (**b**) the Mises stress of No.4 bionic sucker; (**c**) the Mises stress of standard sucker.

**Figure 11 biomimetics-09-00623-f011:**
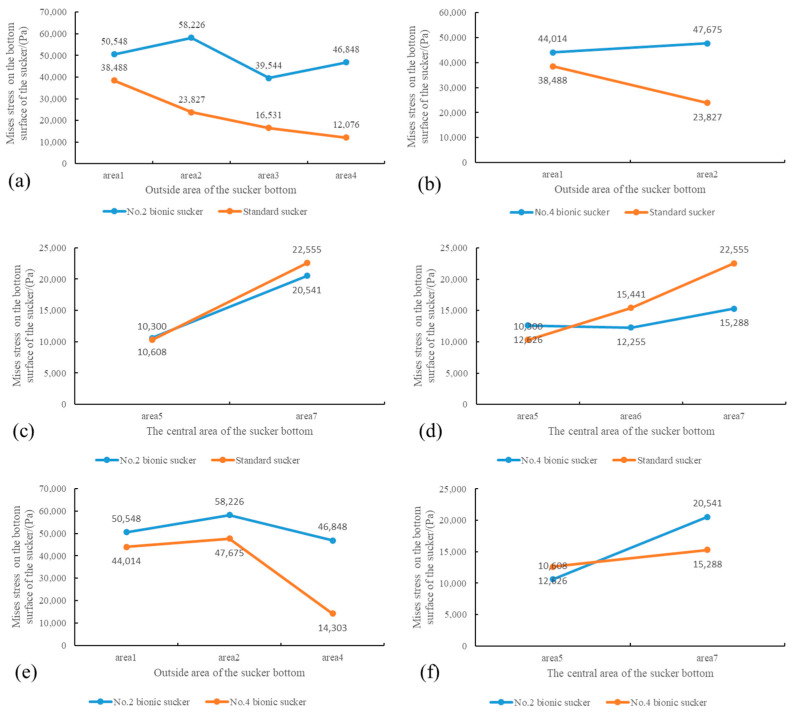
Mises stress comparison of No. 2, No. 4 bionic suckers and standard suckers in different areas on the bottom surface of suckers. (**a**) The Mises stress of No.2 bionic sucker and standard sucker on the outside area of the sucker bottom; (**b**) The Mises stress of No.4 bionic sucker and standard sucker on the outside area of the sucker bottom; (**c**) The Mises stress of No.2 bionic sucker and standard sucker on the central area of the sucker bottom; (**d**) The Mises stress of No.4 bionic sucker and standard sucker on the central area of the sucker bottom; (**e**) The Mises stress of No.2 and No.4 bionic sucker on the outside area of the sucker bottom; (**f**) The Mises stress of No.2 and No.4 bionic sucker on the central area of the sucker bottom.

**Table 1 biomimetics-09-00623-t001:** Factor levels of the bionic sucker.

	Factor	Sealing Ring Width D/mm	Sealing Ring Number N	Sealing Ring Spacing L/mm
Level				
1		1.5	1	2
2		2.5	2	3
3		3.5	3	4

**Table 2 biomimetics-09-00623-t002:** Test scheme and results analysis of the suckers.

	Factor	Sealing Ring Width D/mm	Sealing Ring Number N	Sealing Ring Spacing L/mm	Adsorption Force F/N
Test Number					
1		1.5	1	2	52
2		1.5	2	3	63
3		1.5	3	4	56.3
4		2.5	1	4	47
5		2.5	2	2	55.8
6		2.5	3	3	52.1
7		3.5	1	3	50
8		3.5	2	4	52.6
9		3.5	3	2	52
10		0	0	0	54.4
ȳ*_Fj_*_1_		57.1	49.7	53.3	
ȳ*_Fj_*_2_		51.6	57.1	55	
ȳ*_Fj_*_3_		51.5	53.5	52	
*R_Fj_*		5.6	7.4	3	
Primary and secondary factors		N, D, L			
Optimal combination		N_2_D_1_L_2_			

**Table 3 biomimetics-09-00623-t003:** Repeated experiments on the optimal combination of bionic suckers.

	Factor	Sealing Ring Width D/mm	Sealing Ring Number N	Sealing Ring Spacing L/mm	Adsorption Force F/N	
Test Number						
1		1.5	2	3	61.9	S_e_ = 0.187 f_e_ = 2
2		1.5	2	3	61.5	
3		1.5	2	3	62.1	

## Data Availability

Data are contained within the article.

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
