# Peer review of "Bionic Design and Adsorption Performance Analysis of Vacuum Suckers"

_biomimetics, 2024, doi:10.3390/biomimetics9100623_

Round 1

Reviewer 1 Report

Comments and Suggestions for Authors

-          Introduction

The introduction of the manuscript provides an insufficient background regarding the limitations of current bionic designs in vacuum suckers. The literature review is outdated, as it only briefly mentions studies from 2012, 2014, and 2019 with no recent citations included, which raises questions about the relevance and originality of the insights presented. Additionally, the choice of abalone as a model for the bionic design is not sufficiently justified, leaving a gap in understanding how this study contributes to the current state of knowledge in this field.

-          Research Design

The research design also has several issues that need to be addressed. The design parameters, particularly the dimensions of the sealing rings and their derivation from the abalone design, are not adequately justified within the manuscript. A more thorough explanation of these choices is necessary. While the manuscript discusses the measurement of adsorption force, it would be helpful to include an illustrative example, such as a force-over-time graph, to better clarify how these results were obtained. The significance of the findings could be strengthened with more repetitions to underline the reproducibility and reliability of the results. Furthermore, the test results are not explained fully transparently; it is unclear how many times the tests were repeated and whether the reported values are mean values derived from a larger set of tests.

While the preparation of sucker samples is described in great detail, this focus detracts from the researched adsorption force, which are not presented with the same level of detail. The reported improvement of 15.8% in adsorption force compared to a standard sucker is not convincingly supported by the data provided. There is a lack of graphical comparisons to substantiate this claim. The finite element analysis appears to be thorough, but the discussion of Mises stress values lacks clarity, leaving the meaning for the performance of the derived sucker unclear. Overall, the presentation of results is not as clear as it could be, and additional graphics derived from the experiments would help enhance the study’s validity.

-          Methods

The methods section is well-explained and provides sufficient detail for others to replicate the study. The steps for designing and testing the bionic sucker are clearly outlined.

-          Conclusion

The conclusion could be enhanced further by more clearly addressing the study's limitations and suggesting directions for future research to round off the findings.

Comments on the Quality of English Language

-          Language

The manuscript contains several grammatical errors and some of the introductory sentences  (e.g. in lines 14 and 31 but also in line 41,) are difficult to read and do not provide a comprehensive introduction to the manuscript and its content and should be revised. The citation style is also inconsistent and needs to be revised to ensure it meets the journal’s standards.

Reviewer 2 Report

Comments and Suggestions for Authors

Authors studied the sealing properties of abalone gastropods to vacuum sucker. The sealing ring structure from the abalone ventral foot and bionic designs suction cups with different parameters. Standard and bionic suction cups were made, and adsorption experiments were carried out. The orthogonal design method was used to study the effects of different structures on adsorption force. However, some the current form of this study cannot be acceptable. Some comments are listed as follows:

1.     Highlight the novelty of applying abalone-inspired bionic design more clearly in the introduction. Although it is discussed, a comparison with existing studies on similar bionic systems (octopus, remora) could provide more context about how this approach differs and advances the field.

2.     Clarify the rationale behind the specific choices of sealing ring dimensions and configurations.  For example, why were these three levels for sealing width and spacing selected (Table 1)?  Providing more background on how these parameters were derived would add to the methodological robustness.

3.     The use of orthogonal experimental design and finite element analysis (FEA) is appropriate for evaluating the performance of the bionic sucker. The range analysis results (Table 2) could benefit from a more detailed statistical interpretation. It would be helpful to include more discussion on the statistical significance of the range differences and why the second sucker design performed better. Discussing possible interactions between factors might also add depth.

4.     In Figure 8, the Mises stress diagrams could benefit from clearer labeling and a more detailed explanation in the text about what the specific stress levels imply for performance.

5.     Consider expanding the Discussion section to relate the findings to real-world applications. While the conclusion highlights the improvements, a section on practical implicationshow this suction cup design might be implemented or its advantages in a specific industrial settingwould add a foothold to the articles innovativeness.

Comments on the Quality of English Language

1.     Minor language improvements are necessary. For example, in the abstract, the sentence "Aiming at the problem that the effect of traditional methods on improving the adsorption performance..." could be rewritten for clarity. Consider: "This study addresses the limited effectiveness of traditional methods in improving vacuum sucker adsorption performance."  Additionally, ensure consistent use of plural vs. singular forms (e.g., "sealing rings" vs.  "sealing ring") throughout.

Round 2

Reviewer 2 Report

Comments and Suggestions for Authors

Authors have revised the manuscript according to the mostly comments. However, the current form of this study cannot be acceptable. Some aspects as listed below:

1.In Fig.9, more details about the finite element model can be given. Such as meshing, conditions?

Round 3

Reviewer 2 Report

Comments and Suggestions for Authors

Authors have revised the manuscript according to the mostly comments. However, some details about the finite element modeling still can be given. Fig 9 have been added. The Mooney-Rivlin 2 Parameter model is selected in the material library, the parameters details can be given. There are many types of rubber materials, the mechanical properties are also different.  Besides, the meshing type and numbers also can be given, which affect the simulation results.
